# Meta-learning Symmetries by Reparameterization

**Allan Zhou, Tom Knowles, Chelsea Finn**
Dept of Computer Science, Stanford University
{ayz,tknowles,cbfinn}@stanford.edu

## Abstract

Many successful deep learning architectures are equivariant to certain transformations in order to conserve parameters and improve generalization: most famously, convolution layers are equivariant to shifts of the input. This approach only works when practitioners know the symmetries of the task and can manually construct an architecture with the corresponding equivariances. Our goal is an approach for learning equivariances from data, without needing to design custom task-specific architectures. We present a method for learning and encoding equivariances into networks by learning corresponding parameter sharing patterns from data. Our method can provably represent equivariance-inducing parameter sharing for any finite group of symmetry transformations. Our experiments suggest that it can automatically learn to encode equivariances to common transformations used in image processing tasks. We provide our experiment code at https://github.com/AllanYangZhou/metalearning-symmetries.

## 1 Introduction

In deep learning, the convolutional neural network (CNN) (LeCun et al., 1998) is a prime example of exploiting *equivariance* to a symmetry transformation to conserve parameters and improve generalization. In image classification (Russakovsky et al., 2015; Krizhevsky et al., 2012) and audio processing (Graves and Jaitly, 2014; Hannun et al., 2014) tasks, we may expect the layers of a deep network to learn feature detectors that are translation equivariant: if we translate the input, the output feature map is also translated. Convolution layers satisfy translation equivariance by definition, and produce remarkable results on these tasks. The success of convolution's "built in" inductive bias suggests that we can similarly exploit *other* equivariances to solve machine learning problems.

However, there are substantial challenges with building in inductive biases. Identifying the correct biases to build in is challenging, and even if we do know the correct biases, it is often difficult to build them into a neural network. Practitioners commonly avoid this issue by "training in" desired equivariances (usually the special case of invariances) using data augmentation. However, data augmentation can be challenging in many problem settings and we would prefer to build the equivariance into the network itself. For example, robotics sim2real transfer approaches train agents that are robust to varying conditions by varying the simulated environment dynamics (Song et al., 2020). But this type of augmentation is not possible once the agent leaves the simulator and is trying to learn or adapt to a new task in the real world. Additionally, building in incorrect biases may actually be detrimental to final performance (Liu et al., 2018b).

In this work we aim for an approach that can automatically learn and encode equivariances into a neural network. This would free practitioners from having to design custom equivariant architectures for each task, and allow them to transfer any learned equivariances to new tasks. Neural network layers can achieve various equivariances through parameter sharing patterns, such as the spatial parameter sharing of standard convolutions. In this paper we *reparameterize* network layers to *learnably* represent sharing patterns. We leverage meta-learning to learn the sharing patterns that help a model generalize on new tasks.

The primary contribution of this paper is an approach to automatically learn equivariance-inducing parameter sharing, instead of using custom designed equivariant architectures. We show theoretically that reparameterization can represent networks equivariant to any finite symmetry group. Our

experiments show that meta-learning can recover various convolutional architectures from data, and learn invariances to common data augmentation transformations.

## 2 RELATED WORK

A number of works have studied designing layers with equivariances to certain transformations such as permutation, rotation, reflection, and scaling (Gens and Domingos, 2014; Cohen and Welling, 2016; Zaheer et al., 2017; Worrall et al., 2017; Cohen et al., 2019; Weiler and Cesa, 2019; Worrall and Welling, 2019). These approaches focus on manually constructing layers analogous to standard convolution, but for other symmetry groups. Rather than building symmetries into the architecture, data augmentation (Beymer and Poggio, 1995; Niyogi et al., 1998) trains a network to satisfy them. Diaconu and Worrall (2019) use a hybrid approach that pre-trains a basis of rotated filters in order to define roto-translation equivariant convolution. Unlike these works, we aim to automatically build in symmetries by acquiring them from data.

Our approach is motivated in part by theoretical work characterizing the nature of equivariant layers for various symmetry groups. In particular, the analysis of our method as learning a certain kind of convolution is inspired by Kondor and Trivedi (2018), who show that under certain conditions all linear equivariant layers are (generalized) convolutions. Shawe-Taylor (1989) and Ravanbakhsh et al. (2017) analyze the relationship between desired symmetries in a layer and symmetries of the weight matrix. Ravanbakhsh et al. (2017) show that we can make a layer equivariant to the permutation representation of any discrete group through a corresponding parameter sharing pattern in the weight matrix. From this perspective, our reparameterization is a way of representing possible parameter sharing patterns, and the training procedure aims to learn the correct parameter sharing pattern that achieves a desired equivariance.

Prior work on automatically learning symmetries include methods for learning invariances in Gaussian processes (van der Wilk et al., 2018) and learning symmetries of physical systems (Greydanus et al., 2019; Cranmer et al., 2020). Another very recent line of work has shown that more general Transformer (Vaswani et al., 2017) style architectures can match or outperform traditional CNNs on image tasks, without baking in translation symmetry (Dosovitskiy et al., 2020). Their results suggest that Transformer architectures can automatically learn symmetries and other inductive biases from data, but typically only with very large training datasets. One can also consider automatic data augmentation strategies (Cubuk et al., 2018; Lorraine et al., 2019) as a way of learning symmetries, though the symmetries are not embedded into the network in a transferable way. Concurrent work by Benton et al. (2020) aims to learn invariances from data by learning distributions over transformations of the *input*, similar to learned data augmentation. Our method aims to learn parameter sharing of the layer *weights* which induces equivariance. Additionally, our objective for learning symmetries is driven directly by generalization error (in a meta-learning framework), while the objective in Benton et al. (2020) adds a regularizer to the training loss to encourage symmetry learning.

Our work is related to neural architecture search (Zoph and Le, 2016; Brock et al., 2017; Liu et al., 2018a; Elsken et al., 2018), which also aims to automate part of the model design process. Although architecture search methods are varied, they are generally not designed to exploit symmetry or learn equivariances. Evolutionary methods for learning both network weights and topology (Stanley and Miikkulainen, 2002; Stanley et al., 2009) are also not motivated by symmetry considerations.

Our method learns to exploit symmetries that are shared by a collection of tasks, a form of meta-learning (Thrun and Pratt, 2012; Schmidhuber, 1987; Bengio et al., 1992; Hochreiter et al., 2001). We extend gradient based meta-learning (Finn et al., 2017; Li et al., 2017; Antoniou et al., 2018) to separately learn parameter sharing patterns (which enforce equivariance) and actual parameter values. Separately representing network weights in terms of a sharing pattern and parameter values is a form of *reparameterization*. Prior work has used weight reparameterization in order to "warp" the loss surface (Lee and Choi, 2018; Flennerhag et al., 2019) and to learn good latent spaces (Rusu et al., 2018) for optimization, rather than to encode equivariance. HyperNetworks (Ha et al., 2016; Schmidhuber, 1992) generate network layer weights using a separate smaller network, which can be viewed as a nonlinear reparameterization, albeit not one that encourages learning equivariances. Modular meta-learning (Alet et al., 2018) is a related technique that aims to achieve combinatorial generalization on new tasks by stacking meta-learned "modules," each of which is a neural network.

This can be seen as parameter sharing by re-using and combining modules, rather than using our layerwise reparameterization.

## 3 PRELIMINARIES

In Sec. 3.1, we review gradient based meta-learning, which underlies our algorithm. Sections 3.2 and 3.3 build up a formal definition of equivariance and *group convolution* (Cohen and Welling, 2016), a generalization of standard convolution which defines equivariant operations for other groups such as rotation and reflection. These concepts are important for a theoretical understanding of our work as a method for learning group convolutions in Sec. 4.2.

### 3.1 GRADIENT BASED META-LEARNING

Our method is a gradient-based meta-learning algorithm that extends MAML (Finn et al., 2017), which we briefly review here. Suppose we have some task distribution $p(\mathcal{T})$, where each task dataset is split into training and validation datasets $\{\mathcal{D}_i^{tr}, \mathcal{D}_i^{val}\}$. For a model with parameters $\theta$, loss $\mathcal{L}$, and learning rate $\alpha$, the "inner loop" updates $\theta$ on the task's training data: $\theta' = \theta - \alpha \nabla_\theta \mathcal{L}(\theta, \mathcal{D}^{tr})$. In the "outer loop," MAML meta-learns a good initialization $\theta$ by minimizing the loss of $\theta'$ on the task's validation data, with updates of the form $\theta \leftarrow \theta - \eta \frac{d}{d\theta} \mathcal{L}(\theta', \mathcal{D}^{val})$. Although MAML focuses on meta-learning the inner loop initialization $\theta$, one can extend this idea to meta-learning other things such as the inner learning rate $\alpha$. In our method, we meta-learn a parameter sharing pattern at each layer that maximizes performance across the task distribution.

### 3.2 GROUPS AND GROUP ACTIONS

Symmetry and equivariance is usually studied in the context of groups and their actions on sets; refer to Dummit and Foote (2004) for more comprehensive coverage. A group $G$ is a set closed under some associative binary operation, where there is an identity element and each element has an inverse. Consider the group $(\mathbb{Z}, +)$ (the set of integers with addition): we can add any two integers to obtain another, each integer has an additive inverse, and 0 is the additive identity.

A group $G$ can act on a set $X$ through some action $\rho : G \to \mathrm{Aut}(X)$ which maps each $g \in G$ to some transformation on $X$. $\rho$ must be a homomorphism, i.e. $\rho(gh) = \rho(g)\rho(h)$ for all $g, h \in G$, and $\mathrm{Aut}(X)$ is the set of automorphisms on $X$ (bijective homomorphisms from $X$ to itself). As a shorthand we write $gx := \rho(g)(x)$ for any $x \in X$. Any group can act on itself by letting $X = G$: for $(\mathbb{Z}, +)$, we define the action $gx = g + x$ for any $g, x \in \mathbb{Z}$.

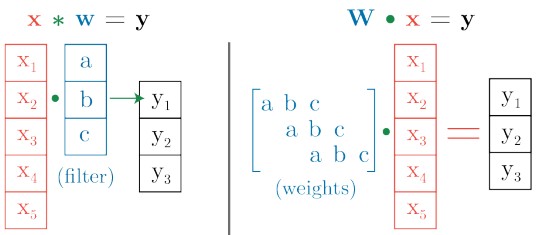

The action of a group $G$ on a vector space $V$ is called a *representation*, which we denote $\pi : G \to GL(V)$. Recall $GL(V)$ is the set of invertible linear maps on $V$. Assume the vectors $v \in V$ are discrete, with components $v[i]$. If we already have $G$'s action on the indices, a natural corresponding representation is defined $(\pi(g)v)[i] := v[g^{-1}i]$. As a concrete example,

Figure 1: **Convolution as translating filters. Left**: Standard 1-D convolution slides a filter $w$ along the length of input $x$. This operation is translation equivariant: translating $x$ will translate $y$. **Right**: Standard convolution is equivalent to a fully connected layer with a parameter sharing pattern: each row contains translated copies of the filter. Other equivariant layers will have their own sharing patterns.

consider the representation of $G = (\mathbb{Z}, +)$ for infinite length vectors. The indices are also integers, so the group is acting on itself as defined above. Then $(\pi(g)v)[i] = v[g^{-1}i] = v[i - g]$ for any $g, i \in \mathbb{Z}$. Hence this representation of $\mathbb{Z}$ *shifts* vectors by translating their indices by $g$ spaces.

### 3.3 EQUIVARIANCE AND CONVOLUTION

A function (like a neural network layer) is equivariant to some transformation if transforming the function's input is the same as transforming its output. To be more precise, we must define what those transformations of the input and output are. Consider a neural network layer $\phi : \mathbb{R}^n \to \mathbb{R}^m$.

Assume we have two representations $\pi_1, \pi_2$ of group $G$ on $\mathbb{R}^n$ and $\mathbb{R}^m$, respectively. For each $g \in G$, $\pi_1(g)$ transforms the input vectors, while $\pi_2(g)$ transforms the output vectors. The layer $\phi$ is *G-equivariant* with respect to these transformations if $\phi(\pi_1(g)v) = \pi_2(g)\phi(v)$, for any $g \in G, v \in \mathbb{R}^n$. If we choose $\pi_2 \equiv id$ we get $\phi(\pi_1(g)v) = \phi(v)$, showing that *invariance is a type of equivariance*.

Deep networks contain many layers, but function composition preserves equivariance. So if we achieve equivariance in each individual layer, the whole network will be equivariant. Pointwise nonlinearities such as ReLU and sigmoid are already equivariant to any permutation of the input and output indices, which includes translation, reflection, and rotation. Hence we are primarily focused on enforcing equivariance in the *linear* layers.

Prior work (Kondor and Trivedi, 2018) has shown that a linear layer $\phi$ is equivariant to the action of some group if and only if it is a group convolution, which generalizes standard convolutions to arbitrary groups. For a specific $G$, we call the corresponding group convolution "$G$-convolution" to distinguish it from standard convolution. Intuitively, $G$-convolution transforms a filter according to each $g \in G$, then computes a dot product between the transformed filter and the input. In standard convolution, the filter transformations correspond to translation (Fig. 1). $G$-equivariant layers convolve an input $v \in \mathbb{R}^n$ with a filter $\psi \in \mathbb{R}^n$. Assume the group $G = \{g_1, \cdots, g_m\}$ is finite:

$$\phi(v)[j] = (v \star \psi)[j] = \sum_i v[i](\pi(g_j)\psi)[i] = \sum_i v[i]\psi[g_j^{-1}i] \tag{1}$$

In this work, we present a method that represents and learns parameter sharing patterns for existing layers, such as fully connected layers. These sharing patterns can force the layer to implement various group convolutions, and hence equivariant layers.

## 4   ENCODING AND LEARNING EQUIVARIANCE

To learn equivariances automatically, our method introduces a flexible representation that can encode possible equivariances, and an algorithm for learning which equivariances to encode. Here we describe this method, which we call Meta-learning Symmetries by Reparameterization (MSR).

### 4.1   LEARNABLE PARAMETER SHARING

As Fig. 1 shows, a fully connected layer can implement standard convolution if its weight matrix is constrained with a particular *sharing pattern*, where each row contains a translated copy of the same underlying filter parameters. This idea generalizes to equivariant layers for other transformations like rotation and reflection, but the sharing pattern depends on the transformation. Since we do not know the sharing pattern a priori, we "reparameterize" fully connected weight matrices to represent them in a general and flexible fashion. A fully connected layer $\phi : \mathbb{R}^n \to \mathbb{R}^m$ with weight matrix $W \in \mathbb{R}^{m \times n}$ is defined for input $x$ by $\phi(x) = Wx$. We can optionally incorporate biases by appending a dimension with value "1" to the input $x$. We factorize $W$ as the product of a "symmetry matrix" $U$ and a vector $v$ of $k$ "filter parameters":

$$\text{vec}(W) = Uv, \quad v \in \mathbb{R}^k, U \in \mathbb{R}^{mn \times k} \tag{2}$$

For fully connected layers, we reshape[1] the vector $\text{vec}(W) \in \mathbb{R}^{mn}$ into a weight matrix $W \in \mathbb{R}^{m \times n}$. Intuitively, $U$ encodes the pattern by which the weights $W$ will "share" the filter parameters $v$. Crucially, we can now separate the problem of learning the sharing pattern (learning $U$) from the problem of learning the filter parameters $v$. In Sec. 4.3, we discuss how to learn $U$ from data.

The symmetry matrix for each layer has $mnk$ entries, which can become too expensive in larger layers. Kronecker factorization is a common approach for approximating a very large matrix with smaller ones (Martens and Grosse, 2015; Park and Oliva, 2019). In Appendix A we describe how we apply Kronecker approximation to Eq. 2, and analyze memory and computation efficiency.

In practice, there are certain equivariances that are expensive to meta-learn, but that we know to be useful: for example, standard 2D convolutions for image data. However, there may be still other symmetries of the data (i.e., rotation, scaling, reflection, etc.) that we still wish to learn

---

[1]We will use the atypical convention that vec stacks matrix entries row-wise, not column-wise.

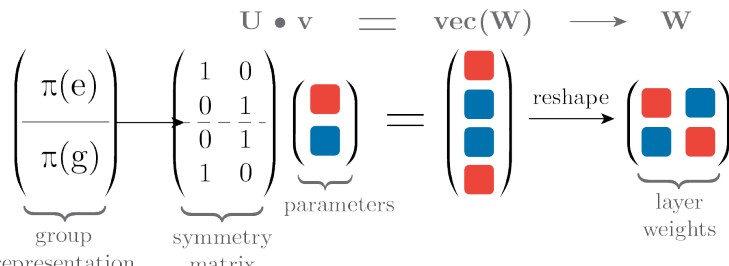

Figure 2: We reparameterize the weights of each layer in terms of a symmetry matrix $U$ that can enforce equivariant sharing patterns of the filter parameters $v$. Here we show a $U$ that enforces permutation equivariance. More technically, the layer implements group convolution on the permutation group $S_2$: $U$'s block submatrices $\pi(e), \pi(g)$ define the action of each permutation on filter $v$. Note that $U$ need not be binary in general.

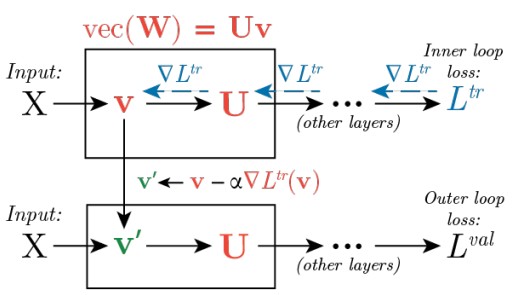

Figure 3: For each task, the inner loop updates the filter parameters $v$ to the task using the inner loop loss. Note that the symmetry matrix $U$ does not change in the inner loop, and is only updated by the outer loop.

---

**Algorithm 1:** MSR: Meta-Training

**input:** $\{\mathcal{T}_j\}_{j=1}^N \sim p(\mathcal{T})$: Meta-training tasks
**input:** $\{\mathbf{U}, \mathbf{v}\}$: Randomly initialized symmetry matrices and filters.
**input:** $\alpha, \eta$: Inner and outer loop step sizes.
**while** *not done* **do**
  sample minibatch $\{\mathcal{T}_i\}_{i=1}^n \sim \{\mathcal{T}_j\}_{j=1}^N$;
  **forall** $\mathcal{T}_i \in \{\mathcal{T}_i\}_{i=1}^n$ **do**
    $\{\mathcal{D}_i^{tr}, \mathcal{D}_i^{val}\} \leftarrow \mathcal{T}_i$;  // task data
    $\boldsymbol{\delta}_i \leftarrow \nabla_{\mathbf{v}}\mathcal{L}(\mathbf{U}, \mathbf{v}, \mathcal{D}_i^{tr})$;
    $\mathbf{v}' \leftarrow \mathbf{v} - \alpha\boldsymbol{\delta}_i$;   // inner step
    /* outer gradient */
    $\mathbf{G}_i \leftarrow \frac{\mathrm{d}}{\mathrm{d}\mathbf{U}}\mathcal{L}(\mathbf{U}, \mathbf{v}', \mathcal{D}_i^{val})$;
  /* outer step */
  $\mathbf{U} \leftarrow \mathbf{U} - \eta \sum_i \mathbf{G_i}$;

---

automatically. This suggests a "hybrid" approach, where we bake-in equivariances we know to be useful, and learn the others. Indeed, we can directly reparameterize a standard convolution layer by reshaping $\text{vec}(W)$ into a convolution filter bank rather than a weight matrix. By doing so we bake in translational equivariance, but we can still learn things like rotation equivariance from data.

## 4.2 PARAMETER SHARING AND GROUP CONVOLUTION

By properly choosing the symmetry matrix $U$ of Eq. 2, we can force the layer to implement arbitrary group convolutions (Eq. 1) by filter $v$. Recall that group convolutions generalize standard convolution to define operations that are equivariant to other transformations, such as rotation. Hence by choosing $U$ properly we can enforce various equivariances, which will be preserved regardless of the value of $v$.

**Proposition 1** *Suppose $G$ is a finite group $\{g_1, \ldots, g_m\}$. There exists a $U^G \in \mathbb{R}^{mn \times n}$ such that for any $v \in \mathbb{R}^n$, the layer with weights $vec(W) = U^G v$ implements $G$-convolution on input $x \in \mathbb{R}^n$. Moreover, with this fixed choice of $U^G$, any $G$-convolution can be represented by a weight matrix $vec(W) = U^G v$ for some $v \in \mathbb{R}^n$.*

Intuitively, $U$ can store the symmetry transformations $\pi(g)$ for each $g \in G$, thus capturing how the filters should transform during $G$-convolution. For example, Fig. 2 shows how $U$ can implement convolution on the permutation group $S_2$. We present a proof in Appendix B.

Subject to having a correct $U^G$, $v$ is precisely the convolution filter in a $G$-convolution. This will motivate the notion of separately learning the convolution filter $v$ and the symmetry structure $U$ in the inner and outer loops of a meta-learning process, respectively.

| Synthetic Problems MSE (lower is better) | | | | | | |
|---|---|---|---|---|---|---|
| | Small train dataset | | | Large train dataset | | |
| Method | $k = 1$ | $k = 2$ | $k = 5$ | $k = 1$ | $k = 2$ | $k = 5$ |
| MAML-FC | $3.4 \pm .60$ | $2.1 \pm .35$ | $1.0 \pm .10$ | $3.4 \pm .49$ | $2.0 \pm .27$ | $1.1 \pm .11$ |
| MAML-LC | $2.9 \pm .53$ | $1.8 \pm .24$ | $.87 \pm .08$ | $2.9 \pm .42$ | $1.6 \pm .23$ | $.89 \pm .08$ |
| MAML-Conv | $\mathbf{.00 \pm .00}$ | $.43 \pm .09$ | $.41 \pm .04$ | $\mathbf{.00 \pm .00}$ | $.53 \pm .08$ | $.49 \pm .04$ |
| MTSR-FC (Ours) | $3.2 \pm .49$ | $1.4 \pm .17$ | $.86 \pm .06$ | $.12 \pm .03$ | $\mathbf{.07 \pm .02}$ | $\mathbf{.07 \pm .01}$ |
| MSR-Joint-FC (Ours) | $.25 \pm .16$ | $\mathbf{.12 \pm .04}$ | $.21 \pm .03$ | $.01 \pm .00$ | $.08 \pm .02$ | $.12 \pm .02$ |
| MSR-FC (Ours) | $.07 \pm .02$ | $\mathbf{.07 \pm .02}$ | $\mathbf{.16 \pm .02}$ | $\mathbf{.00 \pm .00}$ | $.05 \pm .01$ | $.09 \pm .01$ |

Table 1: Meta-test MSE of different methods on synthetic data with (partial) translation symmetry. "Small" vs "large" train dataset refers to the number of examples per training task. Among methods with non-convolutional architectures, MSR-FC is closest to matching actual convolution (MAML-Conv) performance on translation equivariant ($k = 1$) data. On data with less symmetry ($k = 2, 5$), MSR-FC outperforms MAML-Conv and other MAML approaches. MSR-Joint is an ablation of MSR where both $U$ and $v$ of Eq. 2 are updated on task train data, rather than just $v$. MTSR is an ablation of MSR where we train the reparameterization using **m**ulti-**t**ask learning, rather than meta-learning. Results are shown with 95% confidence intervals over test tasks.

### 4.3 META-LEARNING EQUIVARIANCES

Meta-learning generally applies when we want to learn and exploit some shared structure in a distribution of tasks $p(\mathcal{T})$. In this case, we assume the task distribution has some common underlying symmetry: i.e., models trained for each task should satisfy some set of shared equivariances. We extend gradient based meta-learning to automatically learn those equivariances.

Suppose we have an $L$-layer network. We collect each layer's symmetry matrix and filter parameters: $\mathbf{U}, \mathbf{v} \leftarrow \{U^1, \cdots, U^L\}, \{v^1, \cdots, v^L\}$. Since we aim to learn equivariances that are *shared* across $p(\mathcal{T})$, the symmetry matrices should not change with the task. Hence, for any $\mathcal{T}_i \sim p(\mathcal{T})$ the inner loop fixes $\mathbf{U}$ and only updates $\mathbf{v}$ using the task training data:

$$\mathbf{v}' \leftarrow \mathbf{v} - \alpha \nabla_{\mathbf{v}} \mathcal{L}(\mathbf{U}, \mathbf{v}, \mathcal{D}_i^{tr}) \tag{3}$$

where $\mathcal{L}$ is simply the supervised learning loss, and $\alpha$ is the inner loop step size. During meta-training, the outer loop updates $\mathbf{U}$ by computing the loss on the task's validation data using $\mathbf{v}'$:

$$\mathbf{U} \leftarrow \mathbf{U} - \eta \frac{\mathrm{d}}{\mathrm{d}\mathbf{U}} \mathcal{L}(\mathbf{U}, \mathbf{v}', \mathcal{D}_i^{val}) \tag{4}$$

We illustrate the inner and outer loop updates in Fig. 3. Note that in addition to meta-learning the symmetry matrices, we can also still meta-learn the filter initialization $\mathbf{v}$ as in prior work. In practice we also take outer updates averaged over mini-batches of tasks, as we describe in Alg. 1.

After meta-training is complete, we freeze the symmetry matrices $\mathbf{U}$. On a new test task $\mathcal{T}_k \sim p(\mathcal{T})$, we use the inner loop (Eq. 3) to update only the filter $\mathbf{v}$. The frozen $\mathbf{U}$ enforces meta-learned parameter sharing in each layer, which improves generalization by reducing the number of task-specific inner loop parameters. For example, the sharing pattern of standard convolution makes the weight matrix constant along any diagonal, reducing the number of per-task parameters (see Fig. 1).

## 5 CAN WE RECOVER CONVOLUTIONAL STRUCTURE?

We now introduce a series of synthetic meta-learning problems, where each problem contains regression tasks that are guaranteed to have some symmetries, such as translation, rotation, or reflection. We combine meta-learning methods with general architectures *not* designed with these symmetries in mind to see whether each method can automatically *meta-learn* these equivariances.

### 5.1 LEARNING (PARTIAL) TRANSLATION SYMMETRY

Our first batch of synthetic problems contains tasks with translational symmetry: we generate outputs by feeding random input vectors through a 1-D locally connected (LC) layer with filter size 3 and no bias. Each task corresponds to different values of the LC filter, and the meta-learner must minimize mean squared error (MSE) after observing a single input-output pair. For each problem we constrain the LC filter weights with a rank $k \in \{1, 2, 5\}$ factorization, resulting in partial translation symmetry (Elsayed et al., 2020). In the case where rank $k = 1$, the LC layer

is equivalent to convolution (ignoring the biases) and thus generates exactly translation equivariant task data. We apply both MSR and MAML to this problem using a single fully connected layer (MSR-FC and MAML-FC), so these models have no translation equivariance built in and must meta-learn it to solve the tasks efficiently. For comparison, we also train convolutional and locally connected models with MAML (MAML-Conv and MAML-LC). Since MAML-Conv has built in translation equivariance, we expect it to at least perform well on the rank $k = 1$ problem.

We also ran two ablations of MSR that use the same reparameterization (Eq. 2) but vary the training procedure. In MSR-Joint, we allow $U$ and $v$ to be jointly updated in the inner loop, instead of only updating $v$ in the inner loop. Hence MSR-Joint is trained identically to MAML, but with reparameterized weights. MTSR is an ablation of MSR that trains using **m**ulti-**t**ask learning instead of meta-learning. Given data from training task $\mathcal{T}_i$, MTSR jointly optimizes $U$ (shared symmetry matrix) and $v^{(i)}$ (task specific filter parameters) using the MSE loss. For a new test task we freeze the optimized $U$ and optimize a newly initialized filter $v$ using the test task's training data, then evaluate MSE on held out data. Even though the true filter that generates the data has width 3, for MSR and MTSR we initialize the learned filter $v$ to be the same size as the input, per Prop. 1. In principle, these methods should automatically meta-learn that the true filter is sparse, and to ignore the extra dimensions in $v$. Appendix D.1 further explains the experimental setup.

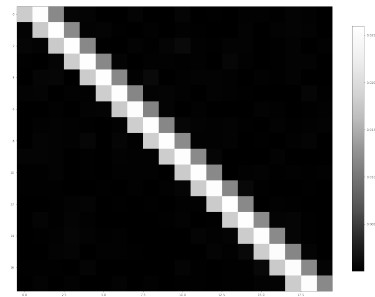

Figure 4: After observing translation equivariant data, MSR enforces convolutional parameter sharing on the weight matrix. An example weight matrix is shown above.

Table 1 shows how each method performs on each of the synthetic problems, with columns denoting the rank $k$ of the problem's data. The "small vs large train dataset" results differ only in that the latter contains 5 or 10 *times* more examples per training task, depending on $k$. On fully translation equivariant data ($k = 1$), MAML-Conv performs best due to its architecture having built in translation equivariance. MSR-FC is the only non-convolutional architecture to perform comparably to MAML-Conv for $k = 1$. Fig. 4 shows that MSR-FC has learned to produce weight matrices with convolutional parameter sharing structure, indicating it has "learned convolution" from the data. Appendix C.1 visualizes the meta-learned $U$, which we find implements convolution as Sec. 4.2 predicted. Meanwhile, MAML-FC and MAML-LC perform significantly worse as they are unable to meta-learn this structure. On partially symmetric data ($k = 2$, $k = 5$), MSR-FC performs well due to its ability to flexibly meta-learn even partial symmetries. MAML-Conv performs worse here since the convolution assumption is overly restrictive, while MAML-FC and MAML-LC are not able to meta-learn much structure. MSR-Joint-FC performs comparably to or worse than MSR-FC across the board. Note that following prior work (Li et al., 2017), all methods use meta-learned inner learning rates on parameters that change in the inner loop. For MSR-Joint-FC we observe that the meta-learned inner loop learning rates corresponding to $U$ are significantly smaller than the inner learning rates corresponding to $v$, suggesting that $U$ is changing relatively little in the inner loop (see Appendix Table 4). MTSR-FC performs significantly worse than MSR-FC with small training datasets, but performs comparably with large datasets. This indicates that although our reparameterization can be trained by either multi-task learning or meta-learning, the meta-learning approach (Alg. 1) is more efficient at learning from less data.

## 5.2 Learning equivariance to rotations and flips

We also created synthetic problems with 2-D synthetic image inputs and outputs, in order to study rotation and flip equivariance. We generate task data by passing randomly generated inputs through a single layer E(2)-equivariant steerable CNN (Weiler and Cesa, 2019) configured to be equivariant to combinations of translations, discrete rotations by increments of 45°, and reflections. Hence our synthetic task data contains rotation and reflection in addition to translation symmetry. Each task corresponds to different values of the data-generating network's weights. We apply MSR and MAML to a single *standard* con-

| Rotation/Flip Equivariance MSE | | |
|---|---|---|
| Method | Rot | Rot+Flip |
| MAML-Conv | .504 | .507 |
| MSR-Conv (Ours) | **.004** | **.001** |

Table 2: MSR learns rotation and flip equivariant parameter sharing on top of a standard convolution model, and thus achieves much better generalization error on meta-test tasks compared to MAML on rotation and flip equivariant data.

volution layer, which guarantees translation equivariance.

Each method must still meta-learn rotation and reflection (flip) equivariance from the data. Table 2 shows that MSR easily learns rotation and rotation+reflection equivariance on top of the convolutional model's built in translational equivariance. Appendix C.2 visualizes the filters MSR produces, which we see are rotated and/or flipped versions of the same filter.

## 6 CAN WE LEARN INVARIANCES FROM AUGMENTED DATA?

Practitioners commonly use data augmentation to train their models to have certain invariances. Since invariance is a special case of equivariance, we can also view data augmentation as a way of learning equivariant models. The downside is that we need augmented data for each task. While augmentation is often possible during meta-training, there are many situations where it is impractical at meta-test time. For example, in robotics we may meta-train a robot in simulation and then deploy (meta-test) in the real world, a kind of sim2real transfer strategy (Song et al., 2020). During meta-training we can augment data using the simulated environ-

---
**Algorithm 2:** Augmentation Meta-Training

**input:** $\{\mathcal{T}_i\}_{i=1}^N$: Meta-training tasks
**input:** META-TRAIN: Any meta-learner
**input:** AUGMENT: Data augmenter
**forall** $\mathcal{T}_i \in \{\mathcal{T}_i\}_{i=1}^N$ **do**
    $\{\mathcal{D}_i^{tr}, \mathcal{D}_i^{val}\} \leftarrow \mathcal{T}_i$ ;   // task data split
    $\hat{\mathcal{D}}_i^{val} \leftarrow \text{AUGMENT}(\mathcal{D}^{val})$;
    $\hat{\mathcal{T}}_i \leftarrow \{\mathcal{D}^{tr}, \hat{\mathcal{D}}_i^{val}\}$
META-TRAIN$\left(\{\hat{\mathcal{T}}_i\}_{i=1}^N\right)$

---

ment, but we cannot do the same at meta-test time in the real world. Can we instead use MSR to learn equivariances from data augmentation at training time, and encode those learned equivariances into the network itself? This way, the network would preserve learned equivariances on new meta-test tasks without needing any additional data augmentation.

Alg. 2 describes our approach for meta-learning invariances from data augmentation, which wraps around any meta-learning algorithm using generic data augmentation procedures. Recall that each task is split into training and validation data $\mathcal{T}_i = \{\mathcal{D}_i^{tr}, \mathcal{D}_i^{val}\}$. We use the data augmentation procedure to *only modify the validation data*, producing a new validation dataset $\hat{\mathcal{D}}_i^{val}$ for each task. We re-assemble each modified task $\hat{\mathcal{T}}_i \leftarrow \{\mathcal{D}_i^{tr}, \hat{\mathcal{D}}_i^{val}\}$. So for each task, the meta-learner observes unaugmented training data, but must generalize to augmented validation data. This forces the model to be invariant to the augmentation transforms without actually seeing any augmented training data.

We apply this augmentation strategy to Omniglot (Lake et al., 2015) and MiniImagenet (Vinyals et al., 2016) few shot classification to create the *Aug-Omniglot* and *Aug-MiniImagenet* benchmarks. Our data augmentation function contains a combination of random rotations, flips, and resizes (rescaling), which we only apply to task validation data as described above. The problem is set up analogous to (Finn et al., 2017): for each task, the model must classify images into one of either $5$ or $20$ classes ($n$-way) and receives either $1$ or $5$ examples of each class in the task training data ($k$-shot). Unlike Finn et al. (2017) our Aug-Omniglot and Aug-MiniImagenet benchmarks contain transformed task validation data.

We tried combining Alg. 2 with our MSR method and three other meta-learning algorithms: MAML (Finn et al., 2017), ANIL (Raghu et al., 2019), and Prototypical Networks (ProtoNets) (Snell et al., 2017). While the latter three methods all have the potential to learn equivariant features through Alg. 2, we hypothesize that since MSR enforces learned equivariance through its symmetry matrices it should outperform these feature-metalearning methods. We also paired MAML with a model that has built in equivariance to the group D8 ($45°$-increment rotation and reflections) using the E2-CNN library (Weiler and Cesa, 2019). We call this baseline "MAML+D8". Appendix D.3 describes the experimental setup and methods implementations in more detail.

Table 3 shows each method's meta-test accuracies on both benchmarks. Across different settings MSR performs either comparably to the best method, or the best. MAML and ANIL perform similarly to each other, and usually worse than MSR, suggesting that learning equivariant or invariant features is not as helpful as learning equivariant layer structures. ProtoNets perform well on the easier Aug-Omniglot benchmark, but evidently struggle with learning a transformation invariant metric space on the harder Aug-MiniImagenet problems. MSR even outperforms the architecture with built in rotation and reflection symmetry (MAML+D8) across the board. MSR's advantage may be due

| Method | Aug-Omniglot | | | | Aug-MiniImagenet | |
|---|---|---|---|---|---|---|
| | 5 way | | 20 way | | 5 way | |
| | 1-shot | 5-shot | 1-shot | 5-shot | 1-shot | 5-shot |
| MAML | $87.3 \pm 0.5$ | $93.6 \pm 0.3$ | $67.0 \pm 0.4$ | $79.9 \pm 0.3$ | $42.5 \pm 1.1$ | $61.5 \pm 1.0$ |
| MAML (Big) | $89.3 \pm 0.4$ | $94.8 \pm 0.3$ | $69.6 \pm 0.4$ | $83.2 \pm 0.3$ | $37.2 \pm 1.1$ | $63.2 \pm 1.0$ |
| ANIL | $86.4 \pm 0.5$ | $93.2 \pm 0.3$ | $67.5 \pm 3.5$ | $79.8 \pm 0.3$ | $43.0 \pm 1.1$ | $62.3 \pm 1.0$ |
| ProtoNets | $92.9 \pm 0.4$ | $\mathbf{97.4 \pm 0.2}$ | $\mathbf{85.1 \pm 0.3}$ | $\mathbf{94.3 \pm 0.2}$ | $34.6 \pm 0.5$ | $54.5 \pm 0.6$ |
| MAML + D8 | $94.6 \pm 0.4$ | $96.4 \pm 0.3$ | $82.6 \pm 0.3$ | $85.1 \pm 0.3$ | $44.9 \pm 1.2$ | $56.8 \pm 1.1$ |
| MSR (Ours) | $\mathbf{95.3 \pm 0.3}$ | $\mathbf{97.7 \pm 0.2}$ | $84.3 \pm 0.2$ | $92.6 \pm 0.2$ | $\mathbf{45.5 \pm 1.1}$ | $\mathbf{65.2 \pm 1.0}$ |

Table 3: Meta-test accuracies on Aug-Omniglot and Aug-MiniImagenet few-shot classification. These benchmarks test generalization to augmented validation data from un-augmented training data. MSR performs comparably to or better than other methods under this augmented regime. Results are shown with 95% confidence intervals over test tasks.

to the additional presence of scaling transformations in the image data; we are not aware of architectures that build in rotation, reflection, *and* scaling equivariance at the time of writing. Note that MSR's reparameterization increases the number of meta-learned parameters at each layer, so MSR models contain more total parameters than corresponding MAML models. The "MAML (Big)" results show MAML performance with very large models containing more total parameters than the corresponding MSR models. The results show that MSR also outperforms these larger MAML models despite having fewer total parameters.

## 7 DISCUSSION AND FUTURE WORK

We introduce a method for automatically meta-learning equivariances in neural network models, by encoding learned equivariance-inducing parameter sharing patterns in each layer. On new tasks, these sharing patterns reduce the number of task-specific parameters and improve generalization. Our experiments show that this method can improve few-shot generalization on task distributions with shared underlying symmetries. We also introduce a strategy for meta-training invariances into networks using data augmentation, and show that it works well with our method. By encoding equivariances into the network as a parameter sharing pattern, our method has the benefit of preserving learned equivariances on new tasks so it can learn more efficiently.

Machine learning thus far has benefited from exploiting human knowledge of problem symmetries, and we believe this work presents a step towards learning and exploiting symmetries automatically. This work leads to numerous directions for future investigation. In addition to generalization benefits, standard convolution is practical since it exploits the parameter sharing structure to improve computational efficiency, relative to a fully connected layer of the same input/output dimensions. While MSR we can improve computational efficiency by reparameterizing standard convolution layers, it does not exploit learned structure to further optimize its computation. Can we automatically learn or find efficient implementations of these more structured operations? Additionally, MSR is focused on learning *finite* symmetry groups, while approximating infinite ones (e.g., learning $45°$-increment rotation symmetry as an approximation to continuous rotation symmetry). Unfortunately, the number of parameters increases with the resolution of the approximation, so further research would be useful in discovering more scalable methods of approximating and learning continuous symmetries. Finally, our method is best for learning symmetries which are shared across a distribution of tasks. Further research on quickly discovering symmetries which are particular to a single task would make deep learning methods significantly more useful on many difficult real world problems.

ACKNOWLEDGEMENTS

We would like to thank Sam Greydanus, Archit Sharma, and Yiding Jiang for reviewing and critiquing earlier drafts of this paper. This work was supported in part by Google. CF is a CIFAR Fellow.

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

## A APPROXIMATION AND TRACTABILITY

### A.1 FULLY CONNECTED

From Eq. 2 we see that for a layer with $m$ output units, $n$ input units, and $k$ filter parameters the symmetry matrix $U$ has $mnk$ entries. This is too expensive for larger layers, so in practice, we need a factorized reparameterization to reduce memory and compute requirements when $k$ is larger.

For fully connected layers, we use a Kronecker factorization to scalably reparameterize each layer. First, we assume that the filter parameters $v \in \mathbb{R}^{kl}$ can be arranged in a matrix $V \in \mathbb{R}^{k \times l}$. Then we reparameterize each layer's weight matrix $W$ similar to Eq. 2, but assume the symmetry matrix is the Kronecker product of two smaller matrices:

$$\text{vec}(W) = (U_1 \otimes U_2)\text{vec}(V), \quad U_1 \in \mathbb{R}^{n \times l}, U_2 \in \mathbb{R}^{m \times k} \tag{5}$$

Since we only store the two Kronecker factors $U_1$ and $U_2$, we reduce the memory requirements of $U$ from $mnkl$ to $mk + nl$. In our experiments we generally choose $V \in \mathbb{R}^{m \times n}$ so $U_1 \in \mathbb{R}^{n \times n}$ and $U_2 \in \mathbb{R}^{m \times m}$. Then the actual memory cost of each reparameterized layer (including both $U$ and $v$) is $m^2 + n^2 + mn$, compared to $mn$ for a standard fully connected layer. So in the case where $m \approx n$, MSR increases memory cost by roughly a constant factor of 3.

After approximation MSR also increases computation time (forward and backward passes) by roughly a constant factor of 3 compared to MAML. A standard fully connected layer requires a single matrix-matrix multiply $Y = WX$ in the forward pass (here $Y$ and $X$ are *matrices* since inputs and outputs are in *batches*). Applying the Kronecker-vec trick to Eq. 5 gives:

$$W = U_1 V U_2^T \iff \text{vec}(W) = (U_1 \otimes U_2)\text{vec}(V) \tag{6}$$

So rather than actually forming the (possibly large) symmetry matrix $U_1 \otimes U_2$, we can directly construct $W$ simply using 2 additional matrix-matrix multiplies $W = U_1 V U_2^T$. Again assuming $V \in \mathbb{R}^{m \times n}$ and $m \approx n$, each matrix in the preceding expression is approximately the same size as $W$.

### A.2 2D CONVOLUTION

When reparameterizing 2-D convolutions, we need to produce a filter (a rank-4 tensor $W \in \mathbb{R}^{C_o \times C_i \times H \times W}$). We assume the filter parameters are stored in a rank 3 tensor $V \in \mathbb{R}^{p \times q \times s}$, and factorize the symmetry matrix $U$ into three separate matrices $U_1 \in \mathbb{R}^{C_o \times p}, U_2 \in \mathbb{R}^{C_i \times q}$ and $U_3 \in \mathbb{R}^{HW \times s}$. A similar Kronecker product approximation gives:

$$\tilde{W} = V \times_1 U_1 \times_2 U_2 \times_3 U_3, \quad \tilde{W} \in \mathbb{R}^{C_o \times C_i \times HW} \tag{7}$$

$$W = \text{reshape}(\tilde{W}), \quad W \in \mathbb{R}^{C_i \times C_i \times H \times W} \tag{8}$$

where $\times_n$ represents $n$-mode tensor multiplication (Kolda and Bader, 2009). Just as in the fully connected case, this convolution reparameterization is equivalent to a Kronecker factorization of the symmetry matrix $U$.

An analysis of the memory and computation requirements of reparameterized convolution layers proceeds similarly to the above analysis for the fully connected case. As we describe below, in our augmented experiments using convolutional models each MSR outer step takes roughly $30\% - 40\%$ longer than a MAML outer step.

In practice, for any experiment where we reparameterize a standard 2-D convolution with weights $W \in \mathbb{R}^{C_o \times C_i \times H \times W}$, we choose $p = C_o, q = C_i$, and $s = HW$. Equivalently, we choose $V \in \mathbb{R}^{C_o \times C_i \times HW}$. Although not necessary, this choice conveniently makes the matrices $U_1, U_2$ and $U_3$ into square matrices, which we can initialize to identity matrices at the start of meta-learning.

## B PROOF OF PROPOSITION 1

To show the connection with the existing literature, we first present a slightly generalised definition of $G$-convolution that is more common in the existing literature. We instead model an input signal

as a function $f : X \to \mathbb{R}$ on some underlying space $X$. We then consider a finite group $G = \{g_1, \ldots, g_n\}$ of symmetries acting transitively on $X$, over which we desire $G$-equivariance. Many (but not all) of the groups discussed in (Weiler and Cesa, 2019) are finite groups of this form.

It is proven by (Kondor and Trivedi, 2018) that a function $\phi$ is equivariant to $G$ if and only if it is a $G$-convolution on this space. In the domain of finite groups, we can consider a slight simplification of this notion: a finite "$G$ cross-correlation" of $f$ with a filter $\psi : X \to \mathbb{R}$. This is defined by (Cohen and Welling, 2016) as:

$$[\phi(f)](g) = (f \star \psi)(g) = \sum_{x \in X} f(x)\psi(g^{-1}x).^2 \tag{9}$$

We can now connect this notion with the linear layer, as described in our paper. First, in order for a fully connected layer's weight matrix $W$ to act on function $f$, we must first assume that $f$ has finite support $\{x_1, \ldots, x_s\}$—i.e. $f(x)$ is only non-zero at these $s$ points within $X$. This means that $f$ can be represented as a "dual" vector $\overline{f} \in \mathbb{R}^s$ given by $\overline{f}_i = f(x_i)$, on which $W$ can act.[3]

We aim to show a certain value of $U^G \in \mathbb{R}^{ns \times s}$ allows arbitrary $G$ cross-correlations—and only $G$ cross-correlations—to be represented by fully connected layers with weight matrices of the form

$$\text{vec}(W) = U^G v, \tag{10}$$

where $v \in \mathbb{R}^s$ is any arbitrary vector of appropriate dimension. The reshape specifically gives $W \in \mathbb{R}^{n \times s}$, which transforms the vector $\overline{f} \in \mathbb{R}^s$.

With this in mind, we first use that the action of the group can be represented as a matrix transformation on this vector space, using the matrix representation $\pi$:

$$[\pi(g)\overline{f}]_i = f(g^{-1}x_i) \tag{11}$$

where notably $\pi(g) \in \mathbb{R}^{s \times s}$.

We consider $U^G \in \mathbb{R}^{ns \times s}$, and $v \in \mathbb{R}^s$. Since $v \in \mathbb{R}^s$, we can also treat $v$ as a the "dual" vector of a function $\hat{v} : X \to \mathbb{R}$ with support $\{x_1, \ldots, x_s\}$, described by $\hat{v}(x_i) = v_i$. We can interpret $\hat{v}$ as a convolutional filter, just like $\psi$ in Eq. 9. $W$ then acts on $v$ just as it acts on $\overline{f}$, namely:

$$[\pi(g)v]_i = \hat{v}(g^{-1}x_i). \tag{12}$$

Figure 5: The theoretical convolutional weight symmetry matrix for the group $\langle g \rangle \cong C_4$, where $g$ is a $\frac{N\pi}{2}$-radian rotation of a 3x3 image ($N \in \{0, 1, 2, 3\}$. Notice that the image is flattened into a length 9 vector. The matrix $\pi(g)$ describes the action of a $\frac{N\pi}{2}$ radian rotation on this image.

Now, we define $U^G$ by stacking the matrix representations of $g_i \in G$:

$$U^G = \begin{bmatrix} \pi(g_1) \\ \vdots \\ \pi(g_n) \end{bmatrix} \tag{13}$$

which implies the following value of $W$ :

$$W = \text{reshape}(U^G v) = \text{reshape}\left( \begin{bmatrix} | \\ \pi(g_1)v \\ | \\ \vdots \\ | \\ \pi(g_n)v \\ | \end{bmatrix} \right) = \begin{bmatrix} - & \pi(g_1)v & - \\ & \vdots & \\ - & \pi(g_n)v & - \end{bmatrix} \tag{14}$$

---

[2]This definition avoids notions such as lifting of $X$ to $G$ and the possibility of more general group representations, for the sake of simplicity. We recommend Kondor and Trivedi (2018) for a more complete theory of $G$-convolutions.

[3]This is using the natural linear algebraic dual of the free vector space on $\{x_1, \ldots, x_s\}$.

| Meta-learned LRs on synthetic problems | | | | | | |
|---|---|---|---|---|---|---|
| Variable | Small train dataset | | | Large train dataset | | |
| | $k = 1$ | $k = 2$ | $k = 5$ | $k = 1$ | $k = 2$ | $k = 5$ |
| $U$ | $-0.017$ | $-0.011$ | $-0.052$ | $-0.009$ | $-0.021$ | $-0.039$ |
| $v$ | $+0.241$ | $+0.326$ | $+0.401$ | $+0.241$ | $+0.307$ | $+0.312$ |

Table 4: In the ablation "MSR-Joint-FC" of Sec. 4 we jointly updated $U$ and $v$ in the inner loop with meta-learned inner loop learning rates for each. This is in contrast with standard MSR, where only $v$ is updated in the inner loop (also with a meta-learned learning rate), and $U$ is only updated in the outer loop. The inner learning rates were initialized at 0.02 for all variables. The table shows the inner loop learning rates at the end of training. The relative magnitudes suggest that $v$ is being updated significantly more than $U$ in the inner loop.

This then grants that the output of the fully connected layer with weights $W$ is:

$$(W\overline{f})_i = \sum_{j=1}^{s} (\pi(g_i)v)_j \overline{f}_j. \tag{15}$$

Using that $f$ has finite support $\{x_1, \ldots, x_s\}$, and that $(\pi(g_i)v)_j = \hat{v}(g_i^{-1}x_j)$, we have that:

$$(W\overline{f})_i = \sum_{j=1}^{s} \hat{v}(g_i^{-1}x_j)f(x_j) = \sum_{x \in X} \hat{v}(g_i^{-1}x)f(x). \tag{16}$$

Lastly, we can interpret $W_G\overline{f}$ as a function $\phi^G(f)$ mapping each $g_i \in G$ to its $i^{\text{th}}$ component:

$$[\phi^G(f)](g_i) = (W\overline{f})_i = \sum_{x \in X} \hat{v}(g_i^{-1}x)f(x) \tag{17}$$

which is precisely the cross-correlation as described in Eq. 9 with filter $\psi = \hat{v}$. This implies that $\phi^G$ must be equivariant with respect to $G$. Moreover, all such $G$-equivariant functions are $G$ cross-correlations parameterized by $v$, so with $U^G$ fixed as in Eq.-13, we have that $W = U^G v$ can represent all $G$-equivariant functions.

This means that if $v$ is chosen to have the same dimension as the input, and the weight symmetry matrix is sufficiently large, any equivariance to a finite group can be meta-learned using this approach. Moreover, in this case the symmetry matrix has a very natural and interpretable structure, containing a representation of the group in block submatrices—this structure is seen in practice in our synthetic experiments. Lastly, notice that $v$ corresponds (dually) to the convolutional filter, justifying the notion that we learn the convolutional filter in the inner loop, and the group action in the outer group.

In the above proof, we've used the original definition of group convolution (Cohen and Welling, 2016) for the sake of simplicity. It is useful to note that a slight generalization of the proof applies for more general equivariance between representations, as defined in equation (3.3)—(i.e. the case when $\pi(g)$ is an arbitrary linear transformation, and not necessarily of the form $\pi(g)f(x) = f(g^{-1}x)$.) This is subject to a unitarity condition on the group representation (Worrall and Welling, 2019).

Without any modification to the method, arbitrary linear approximations to group convolution can be learnt when the representation is not a permutation of the indices. For example, non axis-aligned rotations can be easily approximated through both bilinear and bicubic interpolation, whereby the value of a pixel $x$ after rotation is a linear interpolation of the 4 or 16 pixels nearest to the "true" value of this pixel before rotation $g^{-1}x$. Practically, this allows us to approximate equivariance to 45 degree rotations of 2D images, for which there don't exist representations of the form in Eq. 12.

## C   FURTHER SYNTHETIC EXPERIMENT RESULTS

### C.1   VISUALIZING TRANSLATION EQUIVARIANT SYMMETRY MATRICES

Fig. 6 visualizes the actual symmetry matrix $U$ that MSR-FC meta-learns from translation equivariant data. Each column is one of the submatrices $\pi(i)$ corresponding to the action of the discrete

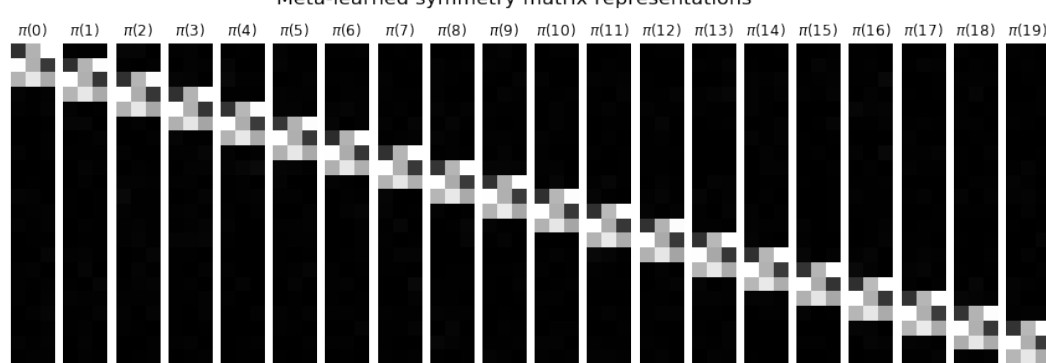

Figure 6: The submatrices of the meta-learned symmetry matrix of MSR-FC on the translation equivariant problem (Sec. 5.1). Intensity corresponds to each entry's absolute value. We see that the symmetry matrix has been meta-learned to implement standard convolution: each $\pi(i)$ translates the size filter $v \in \mathbb{R}^3$ by $i$ spaces. Note that in actuality the submatrices are stacked on top of each other in $U$ as in Eq. 13, but we display them side-by-side for visualization.

| Synthetic problem data quantity | | | | | |
|---|---|---|---|---|---|
| | $k=1$ | $k=2$ | $k=5$ | Rot | Rot+flip |
| No. train tasks | 400 | 800 | 800 | 8000 | 8000 |
| No. test tasks | 100 | 200 | 200 | 2000 | 2000 |
| Examples/train task (Small) | 2 | 2 | 4 | 20 | 20 |
| Examples/train task (Large) | 20 | 20 | 20 | - | - |
| Train examples/test task | 1 | 1 | 1 | 1 | 1 |

Table 5: The amount of training and test data provided to each method in the synthetic experiments of Table 1 and Table 2. The last row indicates that on the test tasks,, all methods were expected to solve each problem using a single example from that task.

translation group element $i \in \mathbb{Z}$ on the filter $v$. In other words, MSR automatically meta-learned $U$ to contain these submatrices $\pi(i)$ such that each $\pi(i)$ translates the filter by $i$ spaces, effectively meta-learning standard convolution! In the actual symmetry matrix the submatrices are stacked on top of each other as in Eq. 13, but we display each submatrix side-by-side for easy visualization. The figure is also cropped for space: there are a total of 68 submatrices but we show only the first 20, and each submatrix is cropped from $70 \times 3$ to $22 \times 3$.

### C.2 VISUALIZING ROTATION AND FLIP EQUIVARIANT FILTERS

In Sec. 5.2 we ran three experiments reparameterizing convolution layers to meta-learn $90°$ rotation, $45°$ rotation, and $45°$ rotation+flip equivariance, respectively. Figure 7 shows that MSR produces rotated and flipped versions of filters in order to make the convolution layers equivariant to the corresponding rotation or flip transformations.

## D    EXPERIMENTAL DETAILS

Throughout this work we implemented all gradient based meta-learning algorithms in PyTorch using the Higher (Grefenstette et al., 2019) library.

### D.1    TRANSLATION SYMMETRY SYNTHETIC PROBLEMS

For the (partial) translation symmetry problems we generated regression data using a single locally connected layer. Each task corresponds to different weights of the data generating network, whose entries we sample independently from a standard normal distribution. For rank $k$ locally connected filters we sampled $k$ width-3 filters and then set the filter value at each spatial location to be a random

linear combination of those $k$ filters. Table 5 shows how many distinct training and test tasks we generated data for. For each particular task, we generated data points by randomly sampling the entries of the input vector from a standard normal distribution, passing the input vector into the data generating network, and saving the input and output as a pair.

**MSR and MAML training**: During meta-training we trained each method for $1,000$ outer steps on task batches of size 32, enough for the training loss to converge for every method in every problem. We used the Adam (Kingma and Ba, 2014) optimizer in the outer loop with learning rate .0005. Like most meta-learning methods, MAML and MSR split each task's examples into a support set (task training data) and a query set (task validation data). On training tasks MAML and MSR used 3 SGD steps on the support data before computing the meta-training objective on the query data, while using 9 SGD steps on the support data of test tasks. We also used meta-learned per-layer learning rates initialized to $0.02$. At meta-test time we evaluated average performance and error bars on held-out tasks.

**MTSR training**: We reparameterize fully connected layers into symmetry matrix $U$ and filter $v$, similar to MSR. MTSR maintains single shared $U$, but initializes a separate filter $v^{(i)}$ for each training task $\mathcal{T}_i$. Given example data $\mathcal{D}_i$ from $\mathcal{T}_i$, we jointly optimize $\{U, v^{(i)}\}$ using the loss $\mathcal{L}(U, v^{(i)}, \mathcal{D}_i)$. In practice each update step updates $U$ and all $\{v^{(i)}\}$ in parallel using the full batch of training tasks. Given a test task we initialize a new filter $v$ alongside our already trained $U$. We then update $v$ on training examples from the test task before evaluating on held out examples from the test task. We use $500$ gradient steps for each task at both training and test time, again using the Adam optimizer with learning rate $0.001$.

We ran all experiments on a single machine with a single NVidia RTX 2080Ti GPU. Our MSR-FC experiments took about $9.5$ (outer loop) steps per second, while our MSR-Conv experiments took about $2.8$ (outer loop) steps per second.

### D.2  ROTATION+FLIP SYMMETRY SYNTHETIC PROBLEMS

The setup of the rotation and rotation+flip symmetry problems is very similar to that of the translation symmetry problems. Here we generated regression data using a single E(2)-steerable (Weiler and Cesa, 2019) layer. Each task again corresponds to a particular setting of the weights of this data generating network, whose entries are sampled from a standard normal distribution for each task. We generate examples for each task similarly to above, and Table 5 shows the quantity of data available for training and test tasks.

MAML and MSR training setups here are similar to the translation setups, but we reparameterize the filter of a standard convolution layer to build in translation symmetry and focus on learning rotation/flip symmetry. Unlike the translation experiments, here we use 1 SGD step in the inner loop for both train and test tasks, and initialize the learned learning rates to $0.1$.

### D.3  AUGMENTATION EXPERIMENTS

To create Aug-Omniglot and Aug-MiniImagenet, we extended the Omniglot and MiniImagenet benchmarks from TorchMeta (Deleu et al., 2019). Each task in these benchmarks is split into support (train) and query (validation) datasets. For the augmented benchmarks we applied data augmentation to only the query dataset of each task, which consisted of randomly resized crops, reflections, and rotations by up to $30°$. Using the torchvision library, the augmentation function is:

```
# Data augmentation applied to ONLY the query set.
size = 28  # Omniglot image size. 84 for MiniImagenet.
augment_fn = Compose(
    RandomResizedCrop(28, scale=(0.8, 1.0)),
    RandomVerticalFlip(p=0.5),
    RandomHorizontalFlip(p=0.5),
    RandomRotation(30, resample=Image.BILINEAR),
)
```

For the augmented Omniglot and MiniImagenet 1-shot experiments, MAML used exactly the same convolutional architecture (same number of layers, number of channels, filter sizes, etc.) as prior

work on Omniglot and MiniImagenet (Vinyals et al., 2016; Finn et al., 2017). For MSR we reparameterize each layer's weight matrix or convolutional filter using the Kronecker approximation (Appendix A) such that the reparameterized layer has the same number of input and output neurons as the corresponding layer in the MAML model.

For MiniImagenet 5-shot, we experimented with increasing architecture size via more channels and/or larger filters, which yielded better accuracies on meta-validation tasks. For MSR, MAML, and ANIL we increased the number of output channels from 32 to 128 and increased the kernel size from 3 to 5 in the first 3 convolution layers. We then inserted a $1 \times 1$ convolution layer with 64 output channels right before the linear output layer. For the ProtoNet architecture we similarly increased the output channels at each layer from 32 to 128, but found that keeping the kernel size at 3 worked best.

For "MAML (Big)" experiments we increased the architecture size of the MAML model to exceed the number of meta-parameters (symmetry matrices + filter parameters) in the corresponding MSR model. For MiniImagenet 5-Shot we inserted an additional linear layer with 3840 output units before the final linear layer. For MiniImagenet 1-Shot we increased the number of output channels at each of the 3 convolution layers from 32 to 64, then inserted an additional linear layer with 1920 output units before the final linear layer. For the Omniglot experiments we increased the number of output channels at each of the 3 convolution layers to 150.

For all experiments and gradient based methods we trained for $60,000$ (outer) steps using the Adam optimizer with learning rate .0005 for MiniImagenet 5-shot and .001 for all other experiments. In the inner loop we used SGD with meta-learned per-layer learning rates initialized to $0.4$ for Omniglot and .05 for MiniImagenet. We meta-trained using a single inner loop step in all experiments, and used 3 inner loop steps at meta-test time. Although MAML originally meta-trained with 5 inner loop steps on MiniImagenet, we found that this destabilized meta-training on our augmented version. We hypothesize that this is due to the discrepancy between support and query data in our augmented problems. During meta-training we used a task batch size of 32 for Omniglot and 10 for MiniImagenet. At meta-test time we evaluated average performance and error bars using 1000 held-out meta-test tasks.

We ran all experiments on a machine with a single NVidia Titan RTX GPU. For our Aug-Omniglot, we ran two experiments at simultaneously on the same machine, which likely slowed each invididual experiment down. Our MSR method took about $0.6$ steps per second, whereas the MAML baseline took about $0.86$ steps per second. For Aug-Miniimagenet we ran one experiment per machine. MSR took 4.2 steps per second, while MAML took 5.6 steps per second on these experiments.

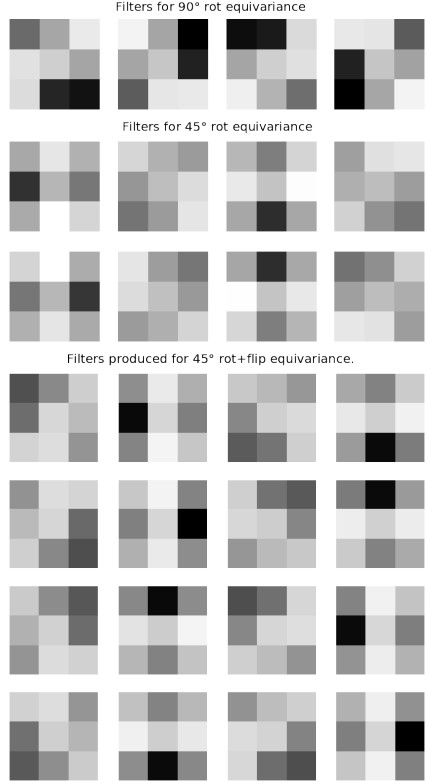

Figure 7: MSR produced convolution filters, after meta-learning $90°$ rotation, $45°$ rotation, and $45°$ rotation+flip equivariance in the Sec. 5.2 experiments. Notice that MSR learns to achieve the corresponding equivariance by producing rotated/flipped versions of the same filter.

