# OpenReview forum: "Meta-learning Symmetries by Reparameterization"
_ICLR.cc/2021/Conference — ICLR 2021 Poster_

### Official Review · AnonReviewer2 · 2020-10-28

**Rating:** 5
**Confidence:** 3

**Review:**

### Summary:
The paper presents a meta-learning algorithm to learn/encode equivariance into deep nets. The main idea is to decompose the model parameters into two parts, a spatial sharing pattern, and the trainable weights. When transferring to a new task, the sharing pattern is fixed and only the remaining trainable weights are tuned. They authors motivated this approach stating that data augmentation may not be practical for robotics application which requires training in the real-world. For experiments, they consider a synthetic dataset where they can recover the equivariance and also k-shot classification tasks on datasets augmented with crops, rotations, and reflections.

### Decision:
I recommend a borderline reject for this paper. I have some questions with the claimed contribution, and details in the experimental section that should be answered before I can recommend an acceptance.

### Supporting Arguments:
1.	Overall, I think the paper is interesting and relevant to the community. However, there are some questions that should be addressed.
2.	This paper demonstrates that through their reparametrization, parameter-sharing, the network can be "equivariant to any finite symmetry group". This result is similar to the work by Ravanbakhsh et al., (2017), which they also show the sharing-patterns encode equivariance properties. Due to its relevance, I believe a more careful discussion should be included and not just cited as “theoretical work has characterized the nature of equivariant layers for various symmetry groups”.

3.	For the experiments, I think a necessary to have a baseline that treats both U,v as trainable parameters. I wonder if it is necessary to train U and v separately on train and val; Maybe the performance gain comes just from the fact that U is trainable, i.e., this model architecture benefits learning.

4.	What are the reported +/- in the Tables? Is it the standard deviation over several random initialization runs?

5.	Just to confirm, “test-sets” in Aug-Omniglot and Aug-MiniImagenet are augmented as well? If yes, are there still performance gain without the augmentation?

6.	Are the baselines in Table 3 trained with data-augmentation as well? The paper states “benchmarks are identical to prior work (Finn et al., 2017)”; does that mean without data augmentation? Also, what happens if both train and val sets are augmented for Alg. 2?

7.	There seem to be some learning rate, step sizes, and architecture tuning as described in supplementary materials. How are these hyperparameters searched? What metric is being used to pick them and what ranges were considered?

### Additional feedback:
- Figure 2. Is a little bit blurry?
- Maybe also mention the issue of  data augmentation with real-world data in the introduction? It wasn’t clear to me the challenges with data augmentation until much later in the paper.

---

> ### Author Response · Authors · 2020-11-16
> **Paper updated with suggested baseline**
>
> We thank the reviewer for their thoughtful comments and questions.
>
> > Ravanbakhsh et al [...] a more careful discussion should be included
>
> We have updated the Related Work (Sec 2) to more carefully explain the relation of [1] to our theoretical analysis and motivation. We chose to analyze our reparameterization primarily from the group convolution perspective since it allows us to naturally frame U as encoding the symmetry, and v as the convolution filter. It also illuminates why MSR can approximate equivariances to group actions that cannot be represented exactly as a permutation of indices (which [1] assumes), as evidenced in the 45 degree-increment rotation experiments (Sec 5.2).
>
> Overall, we’d like to clarify that more broadly the contribution of this paper is not to show that parameter sharing can lead to equivariances, but to introduce a method for learning the correct equivariance-inducing parameter sharing patterns.
>
> > I think a necessary to have a baseline that treats both U,v as trainable parameters
>
> We ran the synthetic experiments (Sec 5.1, Table 1) with the proposed meta-learning baseline where both $U, v$ are updated using the task train data. Across the board this baseline performs comparably to or slightly worse than standard MSR. The meta-learned learning rates for $U, v$ also suggest that it is meta-learning to not update $U$ very much (relative to $v$), see Appendix Table 4.
>
> > Reported +/- in the tables
>
> Following prior works, the +/- report the 95% confidence intervals for the accuracy (or MSE) over the test tasks. We’ve updated the captions to clarify this.
>
> > “test-sets” in Aug-Omniglot and Aug-MiniImagenet are augmented as well?
>
> This is correct, the test set during meta-test is augmented as well. We did not observe any improvement on the base un-augmented distribution for any method.
>
> > Are baselines in Table 3 trained with data-augmentation
>
> Yes, all baselines are also trained with the same data augmentation scheme as used for MSR. We simply meant that analogous to the previous benchmark, we have 5/20 image classes and either 1 or 5 examples per class. We thank the reviewer for pointing out this confusing wording, we have updated the paper to correct this.
>
> > mention the issue of data augmentation with real-world data in the introduction
>
> Thank you for the feedback, we’ve included more discussion about challenges of real world data augmentation in the introduction.
>
> >  hyperparameters
>
> We have updated the paper (particularly Appendix D.3) to be more clear about hyperparameter selection for the Aug-Omniglot and Aug-Miniimagenet benchmarks (Sec 6). The only systematic sweep of hyperparameters we performed was to try larger architectures (increasing channel depth and filter sizes) on MiniImagenet 5-shot, which we found improved meta-validation accuracies for all methods.
>
> Otherwise, where possible we tried to use the same hyperparameters (architecture, learning rates, and batch sizes) as prior work on the original Omniglot/MiniImagenet benchmarks, and for MSR we used the same hyperparameters as the MAML baseline (identical learning rates and architectures, except that the weight matrices of MSR are reparameterized). In some cases where training losses were unstable, we decreased the number of inner steps or decreased the learning rate relative to prior work.
>
> [1] Ravanbakhsh, S., Schneider, J. and Poczos, B., 2017. Equivariance through parameter-sharing.
> [2] Finn, C., Abbeel, P. and Levine, S., 2017. Model-agnostic meta-learning for fast adaptation of deep networks. arXiv preprint arXiv:1703.03400.

---

> > ### Comment · AnonReviewer2 · 2020-11-24
> > **Clarification on baseline's augmentation.**
> >
> > Thanks for the detailed response. I have a clarifying question.
> >
> > >Yes, all baselines are also trained with the same data augmentation scheme as used for MSR.
> >
> > Just to make sure, "same data augmentation" means that only $D^{val}$ is augmented for the baselines. For a fair comparison, I think data augmentation should also be applied to baselines' $D^{tr}$. Is there a reason not the augment $D^{tr}$?

---

> > > ### Author Response · Authors · 2020-11-25
> > > **Clarification response**
> > >
> > > Thank you for the question!
> > >
> > > Yes, only $D^{val}$ is augmented/transformed in all methods, while $D^{tr}$ is the original data distribution. It is a fair comparison, since all methods receive the same train and val datasets during meta-training and meta-testing.
> > > The goal of this experiment is to evaluate whether each meta-learning method can learn and preserve invariance to the transformations when learning a new task, without having task training data that shows that invariance. Hence, to test this goal, the $D^{tr}$ that each method receives at meta-test time is not augmented. Because the meta-test $D^{tr}$ data is not augmented, it would be inappropriate to augment the meta-train $D^{tr}$ for any method, as the meta-learner would not learn to adapt with unaugmented data.

---

> > > > ### Comment · AnonReviewer2 · 2020-11-25
> > > > **Not sure if I follow the reasoning**
> > > >
> > > > > The goal of this experiment is to evaluate whether each meta-learning method can learn and preserve invariance to the transformations when learning a new task, without having task training data that shows that invariance.
> > > >
> > > > Thanks for the clarification.
> > > >
> > > > - The above goal isn't clear from the main paper; might want to clarify.
> > > > -  Why is this a reasonable goal?  My concern is that "Alg. 2 describes our approach for meta-learning invariances from data augmentation". This means that a **practitioner knows** which data augmentations to be applied. Therefore, from a practitioners' perspective, one would apply data augmentation to $D^{tr}$ for the baselines as well.

---

> > > > > ### Author Response · Authors · 2020-11-25
> > > > > **Thank you for the feedback**
> > > > >
> > > > > Thank you for the feedback!
> > > > >
> > > > > * We will revise the first paragraph of Section 6 to make this goal more clear.
> > > > >
> > > > > * We imagine two potential practical use cases of methods that meta-learn symmetries from a set of tasks: one where the equivariances are not known by the practitioner but shared by a set of datasets/tasks (illustrated in the experiments in Section 5), and another where the equivariances are known but only feasible to augment during meta-training and not meta-testing, e.g. in a sim2real setting where it is easy to manipulate the data in simulation but not in the real world. The experiments in Section 6 illustrate how such a latter setting might look like.
> > > > >
> > > > > Based on your feedback, we will revise Section 6 of the paper to better convey this point.
> > > > > Given that there are only a few hours until paper revisions close, we won’t be able to revise the text in that short time. But, we will upload a revised paper it once revisions are allowed again.
> > > > >
> > > > > Finally, as the problem/direction studied in this paper are quite new, we welcome any further feedback or discussion.

---

### Official Review · AnonReviewer4 · 2020-10-28
**An insightful paper with promising results**

**Rating:** 9
**Confidence:** 4

**Review:**

In this paper, the authors propose MSR, a parametrization of convolutional kernels that allows for meta-learning symmetries shared between several tasks. Each kernel is represented as a product of a structure matrix and a vector of the kernel weights. The kernel weights are updated during the inner loop.  The structure matrix is updated during the outer loop.

Strengths
1. The paper is interesting and is easy to read. The figures help a lot in understanding the discussed ideas.
2. The authors demonstrate that the proposed method outperforms the baseline meta-learning models. They empirically prove that MSR indeed learns valuable symmetries from the set of tasks and the provided data.
3. The related work as well as the experimental part allow for a clear positioning of the proposed approach. It demonstrates a valuable connection between meta-learning and building equivariant models.

I did not find any major weaknesses in the presented paper.

Questions
1. A matrix $W$ of size $8\times8$ can be reparametrized in several ways. The corresponding vector $v$ can be of size $1, 2, 4, \dots 64$. Do you consider the size of the vector as a hyperparameter? If so, how to choose it?
2. If we consider the case of the exact flip symmetry, then the length of $\text{vec}(W)$ must be even. One half encodes the original weight and the other half encodes the flipped weight. So we end up with a constraint between the shape of the matrix and the structure of the symmetry group. The same argument applies to all other symmetry groups. What happens if the constraint is not satisfied? Can we learn a flip symmetry for $W$ of size $7 \times 2$? How $U$ will look in this case?

I enjoyed reading the paper. It is insightful, well-written, and demonstrates several valuable results both theoretical and experimental.

### Decision
The authors answered all my questions. My decision stays the same.

---

> ### Author Response · Authors · 2020-11-16
> **Regarding the questions:**
>
> We thank the reviewer for their thoughtful questions:
> * The size of filter $v$ can be an adjusted hyperparameter.
>    * A good default, if computationally feasible, is to simply set the filter size to be of the same size as the input. For the synthetic time series experiments (Sec 5.1) the inputs are of size 70, so we used $v$ with size 70. Even though the “actual” filter which produced the data is actually width 3, MSR automatically learns to ignore the extra filter dimensions and learns the width-3 convolution. We have updated Sec 5.1 to state this explicitly.
>   * For the image experiments where the reparameterization produces the weights for a standard 2d conv, suppose the conv layer expects weights of shape (out_channels, in_channels, width, height). Then we choose a $v$ of that same shape. Of course, this is not a constraint and other $v$ shapes are possible if $U$ is designed properly. We have clarified this in Appendix A.2.
> * In the particular case of a 7x2 $W$, one could have $U$ such that the final row of $W$ is all constant zeros, so that functionally we have a layer with 6 output entries and an extra constant zero output entry.

---

### Official Review · AnonReviewer1 · 2020-10-28
**Very interesting approach to learned network equivariance**

**Rating:** 8
**Confidence:** 4

**Review:**

OVERVIEW:
The authors present a meta-learning approach for network equivariance where the key idea is that equivariance to a finite group of transformations can be achieved by identifying the sharing pattern of weights. Their proposition claims that a fully connected layer $\phi: \mathbb{R}^n \rightarrow \mathbb{R}^m$ with weights $W$ can be factorized into a symmetry matrix $U$ and filter parameters $v$ where the symmetry matrix encodes desired group-convolutions: $\text{vec}(W) = U v, \hspace{1em} v \in \mathbb{R}^k, U \in \mathbb{R}^{mn \times k}$. Within the meta-learning framework, they learn both $U$ and $v$ as part of the outer and inner steps respectively. They demonstrate that they are able to recover the translational equivariance baked into traditional convolutions using this approach including the expected symmetry pattern (shown in Fig. 4). They are also able to demonstrate this two more scenarios: (i) for a group of translation, discrete $45^\circ$ rotation and reflection, and (ii) for Augmented-Omniglot and Augmented-MiniImagenet, providing empirical evidence that their proposed approach learns meaningful information for network equivariance.

PROS:
- I really liked the idea of equivariance captured as a symmetry pattern (or weight sharing) and being able to learn it from data using meta-learning. This is very interesting and exciting with the ability to "learn" rather than "hope" that equivariance is learned from appropriately augmented data.
- A proof of Proposition 1 is presented in the Appendix which is a nice contribution in my opinion.
- I liked the visualizations of the symmetry pattern for translation equivariance and the filters for discrete rotation equivariance (in appendix). They are visual evidence of achieving what is expected from the proposed approach.
- The experiment in Section 5.2 is helpful in backing up the claim of "hybrid" equivariance where you are able to learn translations + discrete rotations. This is achievable by appropriate filter design for simpler groups (mostly 2D) like in Harmonic-Nets [Worrall et al] but can get complicated for interesting groups like SO(3) & SE(3).

CONS:
- The biggest concern for me are that the experiments are largely on synthetic data which has been randomly generated (Sections 5.1 and 5.2). The only real data experiment is on Aug-Omniglot and Aug-MiniImagenet for a few-shot classification task. Equivariance has been demonstrated to be effective on real data in two setups that I am aware of: (a) 3D Model classification with spherical convolutions like in Cohen & Welling, [Esteves et al](https://arxiv.org/abs/1711.06721), (b) Azimuth and scale estimation on Google Earth images [Henriques and Vedaldi](http://proceedings.mlr.press/v70/henriques17a.html). Applying the proposed method to one of these experimental setups will be very helpful in convincing the readers of their use on real data.
- Is the proposed approach learning local equivariance (like with harmonic filters) or global equivariance (like with warped convolutions where transforming the image into polar coordinates gives it rotation and scale equivariance)?
- From the proof of Proposition 1, it becomes clear why the restriction to a finite group. The question I have is what is needed to move into more general (continuous) groups like SO(2), SO(3), SE(3)? Will an approximation (into some finite pattern with tiling) be good enough? I think that it is a possible future research direction but I am interested in knowing your thoughts about it.
- The methods the authors compare against are meta-learning methods which makes sense given the broader framework the work is in. However, I would like a comparision with other network equivariance via filter design algorithms. I don't expect better results but some discussion of comparable performance without the handcrafted design or the ability to learn equivariance for groups without a handcrafted filter available will be a huge plus.

REASON FOR RATING:
I like the paper and it makes a very good contribution in an important area. It is however held back by the lack of experiments on real data and comparision with other established equivariance works leading to my current rating.

UPDATE:
I have read the author feedback and other reviews/discussions. I have updated my rating to 8 from 7 reflect it.

---

> ### Author Response · Authors · 2020-11-16
> **Updated to include new comparison**
>
> We are glad that the reviewer appreciated the idea and visualizations from the experiments. Regarding the questions and requests:
>
> > Local vs global equivariance
>
> Unfortunately we’re not entirely familiar with the local vs global equivariance terminology--is this referring to spatial locality within the input, e.g. as in [1]? Assuming this understanding is correct, we’re inclined to say that MSR is learning global rather than local equivariance.
>
> > The question I have is what is needed to move into more general (continuous) groups
>
> We’ve updated the Future Work (Sec 7) with an expanded discussion on the point of continuous groups. We definitely agree the question of continuous groups is an interesting direction for future research. MSR can in some cases approximate continuous groups, to the extent that a discrete and finite group approximates a continuous one (e.g., Sec 5.2’s 45-degree increment rotations). This can naturally be extended to even finer approximations of continuous rotation, but this will increase the size of U, so a more scalable solution is desirable for the continuous case.
>
> > A comparison with network equivariance via filter design
>
> We’ve updated the paper (Sec 6, Table 3) with partial results from a rotation+reflection equivariant architecture (using filters from "E(2)-equivariant networks" [2]) trained with MAML on Aug-MiniImagenet. The results are partial because each MAML outer step with the equivariant filters is significantly (>10x) slower than with regular CNNs, so the results were recorded at 5k out of 60k outer steps. The training process is ongoing, and we expect those results to improve slightly (we will update again once they are finalized, and also add the results on Aug-Omniglot).
>
> Note this baseline does not have scaling equivariance--we are not currently aware of accessible hand-designed filters with simultaneous scaling, rotation, and reflection equivariances. We are happy to compare with such a baseline if one exists, though.
>
> [1] Kanazawa, Angjoo, Sharma, Abhishek, and Jacobs, David. Locally scale-invariant convolutional neural networks.arXiv preprint arXiv:1412.5104, 2014.
>
> [2] Weiler, M. and Cesa, G., 2019. General e(2)-equivariant steerable cnns. In Advances in Neural Information Processing Systems (pp. 14334-14345).

---

### Official Review · AnonReviewer3 · 2020-10-29
**Meta-learning equivariance by layers with a symmetry matrix and filter, and meta-learning invariance from augmented data**

**Rating:** 6
**Confidence:** 4

**Review:**

This work (i) meta-learns equivariances in neural networks by reparameterizing fully connected layers into a symmetry matrix and filter parameters,
and (ii) meta-learns invariances from augmented data.

Strengths:
+ The work implements (i) by a layer and custom inner loop optimizer using the higher-order optimization library [1]
and (ii) by augmenting benchmark datasets [2].
+ The work learns partial translational symmetry, euqivariance to rotation and flips
+ The work performs important ablation studies, such as testing reparameterization with and without meta-learning by using multi-task learning instead.


Weaknesses:
- Mostly toy examples

- Rather than baking-in inductive biases into the network by encoding equivarainces,
other approaches such as the vision Transformer [3] learn inductive biases from data:
learning local, medium, and long range connections, discovering architectures which supersede CNNs, learning filters,
and demonstrating that CNNs are a curve on an attention distance vs. network depth plane.
Adding a reference to this line of work may improve the introduction.

- Minor changes:
Figures 1,2, and 3 may be improved.
Algorithm 2 may be sufficiently described in words.
Typos on page 14 may be fixed.
lines 9-10 should read "chosen to have" and "theoretically meta-learned"
line 32 should read "each $\pi(i)$ translates the filter"


[1] Generalized inner loop meta-learning, Grefenstette et al, 2019.
https://github.com/facebookresearch/higher

[2] Torchmeta: A meta-learning library for PyTorch, Wurfl et al, 2019.
https://github.com/tristandeleu/pytorch-meta

[3] An image is worth 16x16 words: Transformers for images recognition at scale, Dosovitskiy et al, 2020
https://arxiv.org/pdf/2010.11929.pdf

---

> ### Author Response · Authors · 2020-11-16
> **Paper updated with corrections and suggested references**
>
> We thank the reviewer for their suggested corrections--we’ve updated the manuscript to correct all the minor errors and typos. We’ve also updated the Related Work (Sec 2) with a discussion on automatic inductive-bias learning with Transformer-style architectures, as in (Dosovitskiy et al, 2020). We’d welcome any more specific feedback on Figures 1, 2, and 3.

---

### Decision · Program_Chairs · 2021-01-07
**Final Decision**

**Decision:**

Accept (Poster)

**Comment:**

The paper proposes an approach to meta-learning symmetries. While several approaches have recently emerged with similar goals, and sometimes greater convenience and empirical performance, the proposed approach has some interesting characteristics, such as changing properties of the architecture to extrapolate these symmetries. There was a quite a spread of opinions about the paper, the empirical results were not strong, and updates to the paper focused on helpful text additions, but did not substantively improve the evaluation or experiments. Notwithstanding, the paper is conceptually interesting, there are no major flaws, and there is sufficient support for it.